# Effects of episodic slow slip on seismicity and stress near a subduction-zone megathrust

Saeko Kita [1,7 ✉], Heidi Houston [2], Suguru Yabe[3], Sachiko Tanaka[4], Youichi Asano[4], Takuo Shibutani[5] & Naoki Suda[6]

Slow slip phenomena deep in subduction zones reveal cyclic processes downdip of locked megathrusts. Here we analyze seismicity within a subducting oceanic slab, spanning ~50 major deep slow slip with tremor episodes over 17 years. Changes in rate, b-values, and stress orientations of in-slab seismicity are temporally associated with the episodes. Furthermore, although stress orientations in the slab below these slow slips may rotate slightly, in-slab orientations 20–50 km updip from there rotate farther, suggesting that previously-unrecognized transient slow slip occurs on the plate interface updip. We infer that fluid pressure propagates from slab to interface, promoting episodes of slow slip, which break mineral seals, allowing the pressure to propagate tens of km further updip along the interface where it promotes transient slow slips. The proposed methodology, based primarily on in-slab seismicity, may help monitor plate boundary conditions and slow slip phenomena, which can signal the beginning stages of megathrust earthquakes.

[1] Building Research Institute (BRI), National Research and Development Agency, Tsukuba, Ibaraki, Japan. [2] University of Southern California, Los Angeles, CA, USA. [3] National Institute of Advanced Industrial Science and Technology (AIST), Tsukuba, Ibaraki, Japan. [4] National Institute of Earth Science and Disaster Resilience (NIED), Tsukuba, Ibaraki, Japan. [5] DPRI, Kyoto University, Uji, Kyoto, Japan. [6] Hiroshima University, Higashi-Hiroshima, Hiroshima, Japan. [7] Present address: Department of Earth and Planetary Science and Berkeley Seismological Laboratory, University of California, Berkeley, CA, USA. ✉email: kita@kenken.go.jp

Slip on the plate interface has the potential to affect the stress field and seismicity within a subducting slab. Slow slip has been associated with some great megathrust earthquakes during the slip-nucleation and/or stress-accumulation process[1]. Seismicity and fluid within a downgoing ocean slab may also be closely related to the occurrence of slow slip phenomena. Studies have examined the interaction between slow slip and in-slab earthquakes, showing that even slow deformation on the plate interface has the potential to change the stress field[2] and seismicity[3] within the subducting slab. An analysis[2] of the Hikurangi subduction zone, which is generally in downdip tension, found a change in orientation of the least compressive stress $\sigma 3$ of $\sim 15°$, using 14 months of in-slab focal mechanisms associated with four slow slip events of up to M6.8.

In this study, we examine stress changes, seismicity rate variations, and b-value variations relative to the timing of about 50 slow slip events with tremors beneath the Kii Peninsula (Fig. 1a) using 17 years of data, paying special attention to the stress changes in the upper plane of the double seismic zone. We used the Japan Meteorological Agency earthquake catalog ($\sim$150,000 earthquakes), the National Institute of Earth Science and Disaster Resilience (NIED) tremor catalog[7] ($\sim$10,600 1-h tremors), the upper surface of the Philippine Sea Plate estimated by receiver functions[4,5], the NIED P-wave polarity focal mechanism catalog (M > 2.0), and a stress tensor inversion method[8]. We find temporal changes of the in-slab seismicity rate, b-values, and stress orientations associated with the slow slip episodes, and infer a key role for fluid pressure propagation in promoting slow slip.

## Results

**Changes in stress orientations in the subducting slab.** In this study, we focus on large slow slip earthquakes that occur on the plate interface downdip of the locked megathrust. The slip drives weak low-frequency seismic signals called tremor, as well as other weak seismic phenomena. We use the spatio-temporal record of tremor activity as a proxy to identify large slow slip events. We refer to these quasi-characteristic slow earthquakes as "slow slip with tremor" (SST) to emphasize that tremor is driven by slow slip and is not a separate process. We determined times of approximately 50 large SST events at depths of $\sim$30 km beneath the Kii Peninsula from January 2002 to March 2019 using hypocenter information for tremors detected by NIED[7,9,10] (Fig. 1b). In the Kii Peninsula, SST typically have moment magnitudes of approximately 6. The centers of the major tremor episodes were used to define an occurrence time for each SST. The mean recurrence interval of the SST events is 5.6 months, with durations assessed by inspection of the tremor ranging from 3 to 15 days and a mean duration of 7.8 days consistent with Ref. [11] (also see Supplementary Fig. 1). We selected 3410 in-slab events (0 < M < 5.5) from the JMA earthquake catalog between January 2002 and March 2019, based on the normal distance from the inferred plate interface[4,5]. In Fig. 1b, we then categorized slab seismicity and available focal mechanisms relative to the occurrence times of the nearby SST (i.e., 2–3 months before or after the SST time) ("Methods"). The results of the stress tensor inversions are shown in Figs. 2 and 3. The temporal changes in seismicity and b-values relative to the times of the nearest SST are shown in Fig. 4.

Double seismic planes in the Philippine Sea Plate beneath the Kii Peninsula have been proposed previously[15], and are here clearly revealed in Fig. 2b–d. The upper and lower planes have sufficiently different stress regimes that they must be analyzed separately. We focus on the upper plane because we expect that the stress regime in this region is more directly influenced by slip

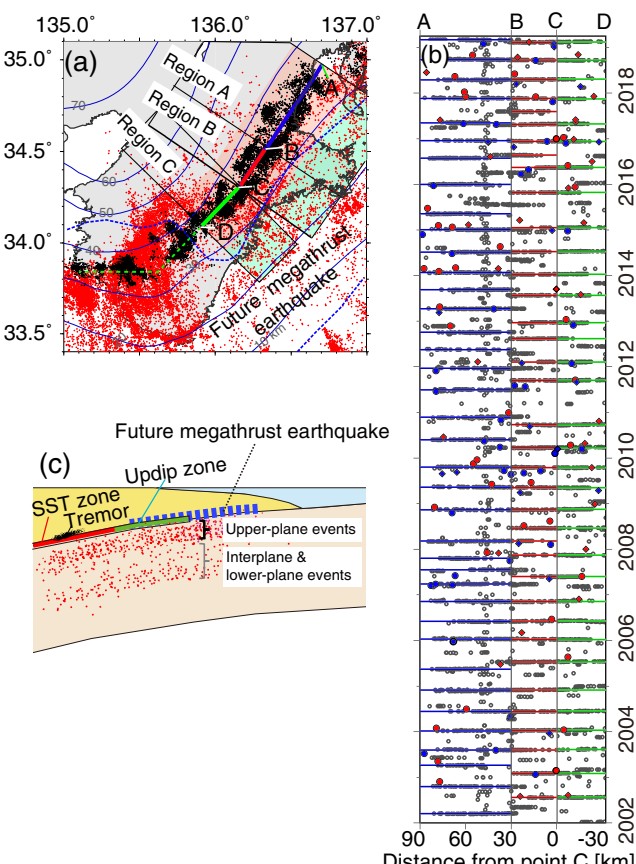

**Fig. 1 Activity of slow slip with tremors (SST) and in-slab earthquakes. a** Tremor (black dots) and slab seismicity (red dots) in Kii Peninsula. Thin black curves show depth contours of the Philippine Sea Plate[4,5]. Blue dashed lines outline regions with a slip-deficit rate greater than 1.5 cm/yr, indicating sites of expected future great megathrust earthquakes[6]. Red and green shaded areas show the tremorgenic zone and the region updip of it. **b** Tremor occurrence times versus distance along the strike of the subducting slab. Times of the major SSTs on different segments are shown by blue (region A, 38 times), red (region B, 37 times), and green (region C, 34 times) lines. The division of regions is shown in (**a**). The distance along the strike is measured from point C in (**a**). Gray circles indicate tremors. Solid blue circles and diamonds show upper-plane slab events with focal mechanisms, beneath the SST zone and the region of updip of it, respectively, that occurred in 2-month time windows before SST. Solid red circles and diamonds show upper-plane slab events with focal mechanisms, beneath the SST zone and the region updip of it, respectively, that occurred in 2-month time windows after SST. **c** Schematic cross-section of the area beneath Kii Peninsula.

at the plate boundary than that in the inter- and lower-plane regions (see Fig. 2i, j, Supplementary Text 1, and Supplementary Fig. 2e, f) due to the proximity to the plate boundary (Fig. 1c).

For detailed analyses, we further divided upper-plane events into events in the SST zone and events updip of the SST zone (Figs. 1 and 2a–d). Figure 2e, f shows the stress tensor inversion results for upper-plane events updip of the SST zone for combined regions A, B, and C using the focal mechanisms for events within 2 months before and after the SST times (see Supplementary Fig. 2a, b and Supplementary Table 1). The best fit $\sigma 1$ before SST is located closer to horizontal than to the dip of the plate interface. After SST, $\sigma 1$ changes by approximately 15 degrees, lying farther above the plate interface dip. The 90% confidence limits for $\sigma 1$ after SST do not overlap those before SST. In contrast, stress inversion of upper-plane events just

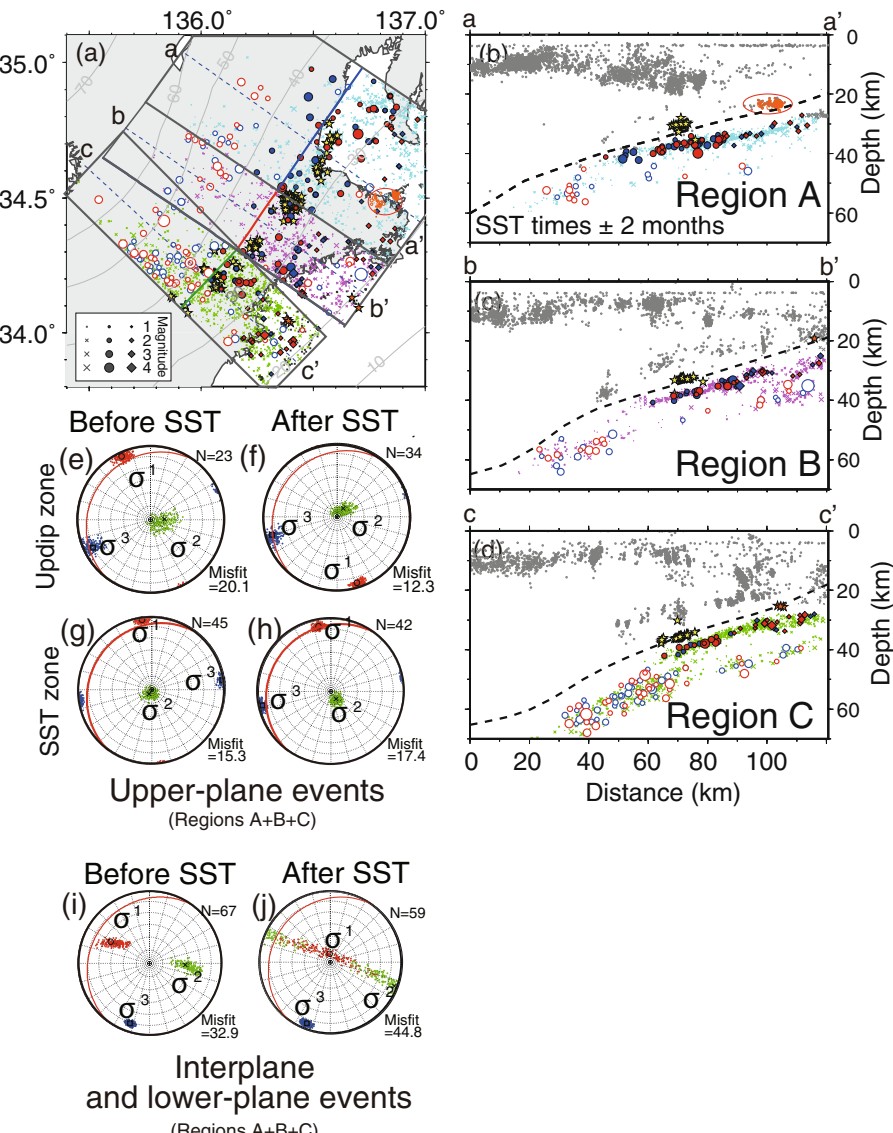

**Fig. 2 Seismicity and results of stress tensor inversions. a–d** Map view and cross-sections of three regions of in-slab seismicity under Kii Peninsula. Explanations of red and blue solid symbols (diamonds and circles) are as in Fig. 1b. Blue and red open circles indicate interplane and lower-plane slab events with focal mechanisms that occurred in 2-month time windows before and after SST times, respectively. Pink, thin blue, and green crosses indicate the seismicity in 2-month time windows before and after the SST times. Yellow stars indicate LFEs located by previous studies[12,13]. Orange stars in (**a–d**) and orange small open dots in (**a**, **b**) indicate repeating earthquakes[14] and a cluster of interplate events, respectively. **e, f** Results of stress tensor inversions for the upper-plane events beneath the region updip of the SST zone for the combined regions A + B + C. Lower-hemisphere projections of σ1, σ2, and σ3, indicated by black circles, crosses, and squares, respectively, of the best-fitting stress tensors before and after SST times (see Supplementary Table 1). The 90% confidence limits are depicted by the distributions of red, green, and blue dots, respectively. Note the shallowing of σ1 after SST. Red lines indicate the orientation of the slab interface. Input data to the inversions are shown in Supplementary Fig. 2. **g, h** Results for the upper-plane events beneath the SST zone. **i, j** Results for the interplane and lower-plane events.

beneath the SST zone (Fig. 2g, h and Supplementary Fig. 2c, d) yields only a small stress change after SST. The best fit σ1 before the SST is horizontal and steepens by ~5° afterward to become parallel to the plate interface dip. Bootstrap analysis in Fig. 3 demonstrates the robustness of the stress changes, particularly for the updip region. The stress ratios R for the results in Fig. 2e–h range between ~0.7 and 0.85.

We also examined the evolution of the stress ratio R from stress tensor inversions, first by using all in-slab events beneath the combined regions A, B, and C (Supplementary Fig. 7). Events were grouped in 1.5 and 2 mo time windows. R appears to reach its maximum about 1.5 mo before the occurrence of slow slip and

then decreases prior to slow slip. R drops from 0.7 to 0.3, and the compression axis becomes less dominant around SST times. However, because we observed a distinct contrast in the stress states between the upper and lower seismic zones, as well as some differences between the SST zone and the region updip of SST, we also analyzed the time evolution of stress for upper-plane events only in these two regions separately (Supplementary Figs. 8–10). Stress ratio R is considered a difficult parameter to resolve in stress inversions[8], and a lack of data may have obscured possible time variations in R. Our analysis finds a consistent variation only for the dip of the σ1 axis in the region updip of SST with a smooth decrease from a peak about 1.5 months prior to slow slip (Supplementary Fig. 10a).

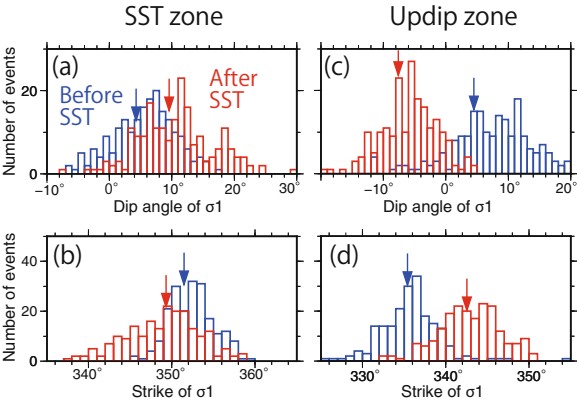

**Fig. 3 Results of 200 inversions of σ1 using bootstrap resampled focal mechanisms for upper-plane events.** Distributions of (**a**) dip and (**b**) strike angles of σ1 beneath the SST zone. Blue and red columns, respectively, show the directions of σ1 of 200 inversions using events before and after SST times. Blue and red arrows, respectively, show the directions of σ1 of the stress inversion using the original dataset before and after SST times. Distributions of (**c**) dip and (**d**) strike angles of σ1 beneath the updip zone.

On the other hand, Ref. [2] obtained the time evolution of the stress ratio from in-slab earthquakes below four shallow slow slip events on the Hikurangi subduction zone. Stress ratios declined significantly for about 6 weeks prior to two of the events, suggesting increasing fluid pressures during those times. The tectonic setting of their study contrasts with ours, as they study fewer but larger slow slip events, at shallower depths, driven by downdip tension rather than downdip compression, and further they do not address the possibility of a double seismic zone with distinct stress states. Despite these differences, their results provide support for our interpretation below that fluid pressures vary over the slow slip cycle in Kii Peninsula.

The plate boundary between the overlying and oceanic plates is likely fully coupled in the region anticipated to generate great megathrust earthquakes, whereas the plate boundary in the SST zone undergoes transient slow slip and is only partially coupled[6,16]. Our observed change in stress orientations suggests that previously-unrecognized transient slow slip occurs on the plate interface 20–50 km updip. This updip region overlaps inferred asperities of anticipated great earthquakes (Fig. 1a–c)[11]. We propose two possibilities to explain the larger stress change we observe updip.

First, larger, but less frequent, slow slip events may occur on the plate interface updip, compared to SST recurrence periods of about 6 months. This appears consistent with reported along-dip trends in Cascadia[17] and Nankai[18] in which size and recurrence intervals of tremor bursts and low-frequency earthquakes increase updip through the SST zone[17], with the degree of plate coupling perhaps controlled by the temperature structure[19]. Alternatively, the same slip on the updip plate interface as in the SST zone could generate more rotation of updip stress axes, if the average effective rigidity there is greater. Higher fluid pressure, and hence lower effective rigidity, in the slab below SST is suggested by previous studies of Vp/Vs[5] and attenuation[20], and plays a key role in our interpretation below.

**Seismicity changes in the subducting slab.** Cumulative numbers of in-slab seismicity versus time relative to SST times in the three regions are shown in Fig. 4a. Increased seismicity rates appear to occur approximately from 2 to 4 weeks before SST times. Cumulative numbers of seismicity versus time relative to SST

times in the combined regions were calculated for three levels of catalog completeness (Mc = 0.5, 0.7, 1.0) (Fig. 4b). Increased seismicity rates clearly occur from about 2 weeks before to a week after SST times. In addition, the b-values, which indicate the relative abundance of small and large earthquakes, of the in-slab earthquakes in the combined regions after the SST were smaller than those before SST (Fig. 4c, Supplementary Text 2, and Supplementary Fig. 4a, b). In summary, b-values are generally larger before the SST times than after, and have a clear peak one to 2 months before the SST times (Fig. 4d). A peak in b-values before the SST times is also observed for the upper-plane seismicity alone (see Supplementary Fig. 5 and Supplementary Text 3). For the evolution of the stress ratio and stress orientations with time, see Supplementary Figs. 7–10 and "Methods".

These changes in seismicity rate and b-values associated with SSTs are consistent with studies of injection seismicity in Switzerland[21] and "geofluid injection" seismicity beneath northeastern Japan[22]. Ref. [21] reported that high pore pressure produced by fluid injection is linked to b-value and that b-value of seismicity in the post-injection period was smaller than that in the co-injection period. Both the b-value and seismicity in the inland crust have been shown to increase (b~2) just after the M9 Tohoku earthquake and to decrease (b~1) within 50 days[22]. Moreover, scaled friction and stress drops for inland events increased within 50 days after the M9 event in that region. High pore pressure and low effective friction at the fault are expected to occur simultaneously in the case of an abundance of fluid. Geofluid injection in the inland crust from the uppermost mantle could have occurred just after the M9 event because the compressional stress field in this region decreased due to the event[22]. After the injection fluid started to disperse, a decrease in b-values and seismicity rate, and an increase in friction on existing faults in the inland crust are thought to have occurred simultaneously[22].

The temporal variation of b-values and seismicity rate in the slab associated with SST, thus appears to be related to fluid pressure redistribution. Because high pore pressure promotes earthquake occurrence, an interpretation of our results, consistent with refs. [21,22], is that high b-value peaks occur due to fluid accumulation in the oceanic slab 1.5 months before the SST times. The decrease in b-values appears to correspond to the start of fluid pressure propagation out of the slab to the plate boundary (Fig. 4d, e). The depth of tremors (30–40 km) is consistent with the location of low P- and S-wave velocities and high Vp/Vs in the oceanic crust[5], which could be a result of fluid migration or fluid pressure propagation from the slab into the plate boundary.

As mentioned, the temporal change of in-slab seismicity suggests fluid accumulation in the slab and propagation of high fluid pressures toward the plate boundary. The low-attenuation zone in the overlying plate[20] above the area with tremor and LFEs in Kii Peninsula[12,13] suggests that the overlying plate there is relatively depleted in fluid. We suggest that fluid from the slab has been trapped beneath the low-attenuation zone, resulting in the generation of transient fluid overpressures in the SST zone. Geological observations of crack-seal quartz veins distributed along the plate-boundary mélange shear zone at the brittle–viscous transition suggest that the veins record repeated brittle failure under fluid overpressures[23]. The recurrence intervals of this process have been estimated to be roughly similar to those of SST[23]. Geochemical analysis of the quartz veins suggests they precipitated from fluids that originated in the serpentinized mantle, possibly slab mantle, and infiltrated to the plate boundary[24]. Our result that in-slab seismicity is still active after the b-values start to decrease (Fig. 4d, e) may imply that the quartz sealing increases high pore pressures near the in-slab seismicity, as well as in the plate boundary.

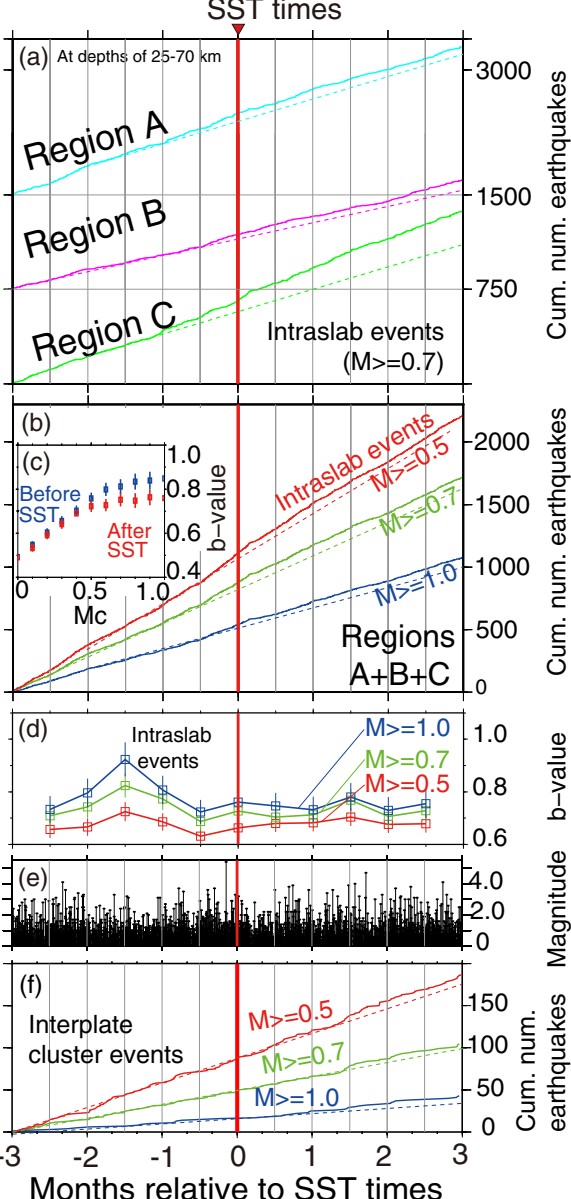

**Fig. 4 Seismicity changes versus time relative to SST times. a** Cumulative numbers of seismicity deeper than 25 km in the three regions shown as solid lines (blue: Region A, pink: Region B and green: Region C) assuming that the lower limit of analysis magnitude, Mc, is 0.7. Note that increased seismicity rates appear to occur ~2–4 weeks before SST times. **b** Cumulative numbers of seismicity in the combined region assuming different levels of completeness (Mc = 0.5, 0.7, 1.0). Increased seismicity rates appear to occur approximately from 2 weeks before to a week after SST times. **c** b-values versus Mc for the combined region. Blue and red dots indicate b-values for events that occurred in 2-month time windows before and after SST times, respectively. Note the decrease in b-values for most values of Mc. An assumed Mc of 0.7 is the preferred value. **d** b-value time variation of in-slab events for the combined region using 1-month moving windows. Red, green, and blue dots indicate the estimated b-values using Mc of 0.5, 0.7, and 1.0, respectively. Note high b-values peak 6 weeks before SST times. **e** Magnitude of events of in-slab events for the combined region. **f** Cumulative numbers of seismicity in the interplate event cluster in the region updip of the SST zone indicated by orange dots in Fig. 2a, b. In (**a**, **b**, **f**), dashed reference lines show the pre-SST trend.

## Discussion

Our results raise the possibility of aseismic slip updip of the SST zone. In general, small- to middle-magnitude repeating earthquakes that occur on the plate boundary breaking the same fault area are evidence of creep on the plate boundary[25]. In the updip zone, two groups of offshore repeating earthquakes (shown in Fig. 2a, c, d as orange stars) occur a few times during our study period. In at least two cases, they occur shortly after SSTs located 30–40 km downdip[14]. In addition, a cluster of earthquakes (shown in Fig. 2a, b as orange small open dots) is located on the plate boundary also about 30 km updip of the SST zone; its seismicity tends to increase ~1.5 months after SST times (Fig. 4f). These observations imply that shortly after SST times, transient slow slip can occur in the updip zone under and offshore Kii Peninsula. Aseismic slip on the plate interface updip of tremor is also consistent with a GPS study, suggesting the occurrence of some slow slip events in the updip zone beneath Kii Peninsula[26] and with other GPS studies implying slip in the region updip of the SST zone in Cascadia[27,28], Shikoku[29] and Tokai regions[26].

Our analysis of in-slab earthquakes relative to SST times suggests that high b-value and seismicity rates in the slab occur before the SST due to high fluid pressure perhaps fluid injection related to dehydration of oceanic crust and/or mantle. Our interpretation is shown schematically in Fig. 5. Before the SST, fluid pressure may propagate from the slab to the plate interface, generating higher pore pressure at the plate boundary. High pore pressure decreases the effective strength of the plate boundary in the SST zone, allowing slip to occur. The slip is associated with a possible small rotation of the stress orientation beneath the SST zone and a larger rotation beneath the region updip of SST. This larger rotation occurs too far from SST to be caused directly by it, but could be related to transient aseismic slip updip of the SST zone. We propose that high pore pressure (and fluid) on the plate boundary disperses after the SST due to rupturing of quartz vein seals. After SST times, fluid pressure on the plate boundary propagates from the SST to the updip zone. Due to the increased pressure, the interface of the updip region becomes less coupled and creep accelerates until sufficient stress is released. Thus, transient aseismic slip also occurs in the updip zone after the SST times as implied by the repeating earthquakes[14] and GPS evidence[26] (Fig. 4f), although these observations are too few to closely resolve the timing. Eventually, stress orientations rotate back to the initial stress state. Different mineralogical, fluid, or temperature conditions in the updip and near-SST regions may produce the observed contrasting effects of SST on plate-boundary coupling in the two regions. Our new approach using focal mechanisms and seismicity in the slab can supplement the recent monitoring efforts for plate-boundary and SST phenomena using only tiltmeters and GPS, which are sparse in remote regions, and subject to bad weather (such as typhoons or heavy rain). The proposed simple methodology could contribute to a better understanding of the nucleation of future megathrust earthquakes.

## Methods

**Stress regime analysis**. To obtain stress orientations and the stress ratio, we performed stress tensor inversions using software from Ref. [8] and focal mechanisms based on P-wave polarities by the National Institute of Earth Science and Disaster Resilience (NIED) catalog (M > 2.0). In the examination of the temporal relationship of in-slab events to SST, we categorized slab seismicity relative to the occurrence times of nearby SST (i.e., 2 or 3 months before or after) (Fig. 1b). We then combined the slab seismicity and events with focal mechanisms based on these relative occurrence times.

Confidence regions are estimated as follows. We assume that errors in the focal mechanisms are Gaussian with an estimated standard deviation of 20°. These random errors are added to the mechanisms, which are then inverted for stress orientations. This process is repeated 200 times. We then estimate 90% confidence

**Fig. 5 Schematic cross-sectional views beneath the Kii Peninsula before and after the occurrence of SST.** The stress state of upper-plane events in the slab changes after SST times. Red and green arrows depict the orientations of maximum compressional stresses. Fluids move from the oceanic plate into the plate boundary, as suggested by the high b-value peak for the upper-plane events 1.5 months before SST times. Increased fluid on the plate boundary reduces the degree of plate coupling between the overlying and subducting plates.

regions based on the 90% of the inversion results that lie closest to the optimal inversion result calculated from the original data.

In the stress tensor inversion analysis, the stress ratio R for focal mechanisms is defined as follows:

$$R = (\sigma 1 - \sigma 2)/(\sigma 1 - \sigma 3) \qquad (1)$$

where σ1, σ2, and σ3 are the maximum compressive stress, intermediate compressive stress, and minimum compressive stress, respectively. $R > 0.5$ indicates that a single-axis compression stress field is dominant in the study area, whereas $R > 0.5$ indicates that a single-axis tension stress field is dominant.

**Stability of stress tensor inversions.** In order to show variations in the σ1 direction, we also applied bootstrap resampling tests to input data for the stress tensor inversions shown in Supplementary Fig. 2a–d. Figure 3 shows the result of stress inversions using 200 time-resampling data, in which the distributions of the strike and dip angles of σ1 for stress tensor inversion are shown. Beneath the SST zone, the distribution of the dip and strike angles after SST times significantly overlaps that before SST times, whereas beneath the updip zone, the distribution of the dip and strike angles after SST times differs from that before SST times.

We also examined the time change of the stress axis of stress tensor inversions for upper-plane events using a 1.5-month time window beneath the combined regions. The results beneath SST and updip zones are shown, respectively, in Supplementary Figs. 8 and 9.

**B-value analysis.** We estimated the time evolution of b-values and seismicity rate relative to the SST occurrence times. For the b-value calculation, we used the following relation:

$$b = \frac{\log_{10} e}{\bar{M} - M_c} \qquad (2)$$

where $\log_{10} e = 0.43429$ and $M_c$ is the complete magnitude (i.e., lower limit of the detected magnitude)[30]. The 95% confidence level for b-value estimations is given by $b/\sqrt{N}$ (N: number of events)[31].

**Spatial classification of in-slab events.** We identified in-slab earthquakes based on normal distance from the inferred plate interface estimated by receiver function analysis[4,5]. The plate model beneath Kii peninsula was estimated using 99 temporary as well as 55 permanent seismic stations, and is broadly similar to a previous model[32] based on only permanent stations. The depth contours of this model[4,5] are clearly consistent with the locations of low-frequency earthquakes[12,13] and repeating earthquakes[14] (Fig. 2b–d). The available seismicity and focal mechanisms in the Kii Peninsula indicate that, as found in many subduction zones[33], the

downgoing slab is organized as a double seismic zone[15,34] (Fig. 2b–d) in which the upper and lower zones exhibit different stress regimes. Therefore, to analyze the stress regimes, we divided in-slab events with focal mechanisms into upper-plane events (Fig. 1b and Supplementary Fig. 2a–d) and other in-slab events (interplane and lower-plane events) (Supplementary Fig. 2e, f) and performed stress tensor inversions (Fig. 2e–j and Supplementary Table 1). We classified in-slab events within 10 km of the plate interface (and at depths of 25–45 km) as upper-plane events, whereas in-slab events farther from the interface were classified as interplane and lower-plane events. The classification of upper-plane events is similar to a previous study beneath northeastern Japan[35].

For more detailed analyses, we further divided upper-plane events into events below the SST zone and events 20–50 km updip of the SST zone (Figs. 1 and 2a–h and Supplementary Fig. 2a–d). We define the SST zone to lie within 20 km from the centers of the tremor sequences which are shown as blue, green, and red lines in Fig. 1a.

## Data availability
The p-wave polarity focal mechanisms were obtained from NIED using the MOWLAS (http://www.hinet.bosai.go.jp/). The hypocenters were obtained from the JMA (https://www.data.jma.go.jp/svd/eqev/data/bulletin/hypo_e.html). The tremor and LFEs catalogs are available at the Slow Earthquake database (http://www-solid.eps.s.u-tokyo.ac.jp/sloweq). Additional data related to this paper may be requested from the authors.

## Code availability
The Matlab code for the stress tensor inversion method adopted in this paper is provided on the web page (https://www.ig.cas.cz/en/stress-inverse).

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

## Acknowledgements

We would like to thank K. Ujiie and M. Otsubo for discussions on quartz veins, and F. Hirose, A. Hasegawa, Y. Ito, N. Uchida, K. Ohta, T. Nishikawa, M. Manga, R. Burgmann, and M. Nakakuki for seismological comments. We also would like to thank T. Igarashi and T. Nishimura for discussions on repeating earthquakes and slip-deficit rate, respectively. This study (S.K.) was supported by JSPS KAKENHI Grant Numbers JP16H06475, JP16K21728, JP17K05637, 18KK0392, 19H04627, 20K04139, 21H05200, 21H05202, and a collaborative research grant from the Disaster Prevention Research Institute (DPRI), Kyoto University 28G-07.

## Author contributions

S.K. and H.H. analyzed the data, made the figures, wrote the manuscript, and interpreted the results. S.Y. contributed to the interpretation of the results. S.T. contributed to the tremor catalog and the interpretation of the results. Y.A. contributed selections of the focal mechanisms and a related discussion. T.S. contributed a discussion of the discontinuities and seismicity. N.S. contributed a discussion of LFEs and tremors.

## Competing interests

The authors declare no competing interests.
