## [Peer Review File · Nature Communications]

REVIEWER COMMENTS

Reviewer #1 (Remarks to the Author):

The authors use earthquake focal mechanisms within the downgoing slab beneath Kii Peninsula (Nankai Trough) to invert for the stress field before, during, and after slow slip events. Tracking the evolution of the stress field and of b-values relative to many slow slip events, they suggest that fluid pore pressure builds up prior to slow slip and is then released during slow slip. The analysis and interpretation seem to be carefully done. I am not an expert in stress inversions, but the authors are using a widely used, previously published method. The authors' results echo those of Warren-Smith et al, *NGeo*, 2019, and are an important contribution even if the results aren't completely novel.

I have several comments below that should be addressed before the paper progresses to publication. I do not think it likely that addressing these comments will significantly impact the conclusions and results of the manuscript.

Comments:

1) Why did the authors decide to redo their own slow slip catalog? Nishimura et al., *JGR*, 2013 (Detection of short-term slow slip events along the Nankai Trough, southwest Japan, using GNSS data) derived a slow slip catalog for Kii peninsula from the geodetic record, which is likely to be more reliable than one derived from indirect observations of slow slip, i.e. a tremor catalog. We know it is the aseismic slip that dominates the moment. Therefore a geodetically detected event is going to be more likely associated with a higher SNR observation of stress changes. Not to mention that while tremor and slow slip are more often than not highly correlated, it is not always the case, e.g. Wech and Bartlow, *GRL*, 2014. I would also mention that the panels in Fig. 1 that show the tectonic tremor in time and space are quite small and difficult to interpret; these however might no longer be necessary if the slow slip catalog from Nishimura et al. is used.

2) I am not a fan of the term episodic tremor and slip (ETS), especially to describe a single slow slip event. It conflates geodetic and seismic observations together, making it difficult to understand what is exactly being analyzed for an uninitiated reader, without adding any meaningful information. I would suggest referring to slow slip as simply slow slip.

3) Although the authors say that most of the identified slow slip events last for less than a week, it is important to present the context of the distribution of slow slip durations. Given that the relative timing of the stress field and b-value changes is plotted with respect to the middle of the slow slip event, it is important to know just how long these events last.

4) On a similar note, why is the relative timing centered on the slow slip events? Plotting relative to a distribution of slow slip durations, the middle of the slow slip event is not a very robust reference to understand what happens before and after slow slip events. I would think it would be much more straightforward to plot everything with respect to the start of the slow slip event. That way everything before that reference time is by definition not impacted by ongoing slip. A separate relative time scale could be created for everything after the end of all slow slip events to explore what happens after slow slip.

5) The authors discuss fluid injection prior to slow slip from the oceanic crust into the plate interface. This does not seem very reasonable given the extremely slow production rates of fluids from metamorphic dehydration reactions (cf. Peacock et al., *Geology*, 2011, High pore pressures and porosity at 35 km depth in the Cascadia subduction zone) and the relatively short recurrence of slow slip in Nankai. Is a more reasonable suggestion that it is cycling of fluid pressures rather than fluid flowing itself?

6) line 189-190 states that eventually the stress rotates back to its original state. Wouldn't the authors be able to estimate the duration of this healing process by extending their stress inversions to seismicity following slow slip?

7) There is quite a bit of text in the supplementary materials. I believe that most of this (if not all) could be

incorporated into the main text and the methods section. Having as much information as possible in one document would be ideal.

8) As noted above, ref. 2 presents similar observations and a similar discussion to this manuscript, yet there is no mention of it past the introduction. To provide further context, I would think there should be further discussion of this paper

9) Many of the lines/symbols in the figures are not defined in the captions, e.g. Fig. 3 panel a: are the dashed lines the pre-slow slip trend? The authors should carefully review the figures and make sure that everything is accounted for in the captions or legends.

Reviewer #2 (Remarks to the Author):

This study clarified spatio-temporal correlation between temporal changes in seismicity rate, b-value, and stress orientation and episodic tremor and slip (ETS) occurring deep in the seismogenic zone using earthquake data observed around the Kii Peninsula in the Nankai Trough, and has discussed possible causes of these variations. The method used to determine the seismic source, seismicity rate, b-value, and stress orientation is well established and the results obtained have been precisely evaluated. The results of this study indicate a possibility of discovering plate slip behavior caused by ETS, which was previously overlooked, and the results of this analysis deserve to be published in Nature communication. On the other hand, in the section discussing the causes of the spatio-temporal changes, there were some jumps in logic and insufficient explanations, probably due to the limited number of words. As mentioned below, my main comments are about the discussion, and these are mainly about descriptions, which the authors could easily correct.

Line 98-99: "The inferred asperities of anticipated great earthquake" cited here was proposed in 2003 for a disaster prevention purpose, which may not be suitable for discussing spatio-temporal changes in ETS and plate slip behavior as shown in Fig. 1. It may be more appropriate to use the results of a recent GNSS-A study on the slip deficit rate.

Line 117-118: In Fig. 3, the solid line showing the cumulative number of earthquakes after ETS is roughly parallel to the dotted line, which indicates the seismicity rate is the same as 1-3 months before ETS, although it is written as "... (seismicity rate?) decrease after ETS times" here.

Line 128-139: The interpretation of the temporal variation of seismicity rate and b-values is mainly discussed based on the observations after the Tohoku earthquake. However, my understanding is that the Tohoku earthquake study referred to here discusses a possibility of fluid migration as an "interpretation" of the observed seismicity rate. Referring to the Tohoku study is only showing an example of a previous study that obtained similar observation, and may not appropriate as a discussion of "a causes" of the observed variation in seismicity rate and b-value. If the Swiss water injection study shows quantitative variations in seismicity and b-value with fluid injection, it would be more appropriate to precisely describe the results of the Swiss water injection study.

Line 148-158: It seems to me the arguments here somehow farfetched. For example, this paragraph mentions that "the formation of quartz veins is generally associated with hydrothermal circulation", therefore, "the quartz vein sealing cycle is related to the in-slab seismicity". To me there are some jumps of logic between the two sentences. More explanation may be necessary. And also, it seems to me that this paragraph mentions that the reference 21 and 22 concluded that "low attenuation layer" = "impermeable layer". Do the reference 21 and 22 conclude so?

Line 196-197: I think this sentence is an exaggeration. The approach by this study is useful for understanding slip phenomena after the events, but it may not suitable for "prediction" because this study's approach seems to require "retrospective" analysis of earthquake data "after" the event occur.

Response to Reviewers

We thank the Reviewers for their helpful comments which have significantly improved the clarity and readability of our manuscript. To summarize, we made a number of edits to improve the clarity, moved one figure and two Text items into the main manuscript from the supplemental materials, and added one figure (Figure S1) to the supplemental materials. We also reorganized and rewrote a paragraph to clarify and strengthen our line of reasoning.

Reviewer #1 (Remarks to the Author):

The authors use earthquake focal mechanisms within the downgoing slab beneath Kii Peninsula (Nankai Trough) to invert for the stress field before, during, and after slow slip events. Tracking the evolution of the stress field and of b-values relative to many slow slip events, they suggest that fluid pore pressure builds up prior to slow slip and is then released during slow slip. The analysis and interpretation seem to be carefully done. I am not an expert in stress inversions, but the authors are using a widely used, previously published method. The authors' results echo those of Warren-Smith et al, NGeo, 2019, and are an important contribution even if the results aren't completely novel.

I have several comments below that should be addressed before the paper progresses to publication. I do not think it likely that addressing these comments will significantly impact the conclusions and results of the manuscript.

Comments:

1) Why did the authors decide to redo their own slow slip catalog? Nishimura et al., JGR, 2013 (Detection of short-term slow slip events along the Nankai Trough, southwest Japan, using GNSS data) derived a slow slip catalog for Kii peninsula from the geodetic record, which is likely to be more reliable than one derived from indirect observations of slow slip, i.e. a tremor catalog. We know it is the aseismic slip that dominates the moment. Therefore a geodetically detected event is going to be more likely associated with a higher SNR observation of stress changes. Not to mention that while tremor and slow slip are more often than not highly correlated, it is not always the case, e.g. Wech and Bartlow, GRL, 2014. I would also mention that the panels in Fig. 1 that show the tectonic tremor in time and space are quite small and difficult to interpret; these however might no longer be necessary if the slow slip catalog from Nishimura et al. is used.

Response:

This is an interesting suggestion. We chose to develop our own catalog for slow slip events for several reasons. The Nishimura catalog referenced by the reviewer only extends through 2012, and contains only about half as many slow slip events as we found by analyzing tremor. Furthermore, although the Nishimura et al. catalog is probably most reliable in terms of moment-release estimates, it has less spatio-temporal resolution than the tremor episodes. In particular, the use of tremor is necessary to better resolve the slow slip segmentation boundaries that are prominent in the Kii deep

slow slip zones (ie., the A, B, and C regions, Fig. 1). These segments are used to classify the in-slab focal mechanisms relative to the timing of the slow slip. They are also needed to define the notable cross-sections displaying clear double seismic zones under Kii (Figs. 2b to 2c) that must be considered when inverting for stress, because the stress states in the upper and lower planes are quite distinct (Figs. 2e-2j).

We have tried to improve the legibility of Figure 1. We feel it an important figure to highlight some characteristics of our data set (for example, in contrast with Ref. 2, Smith-Warren et al, our data include many more SSEs over a much longer time period). Additionally, Figures 1b and 2b-2c illustrate some of the relationships between the different categories of seismicity used in our study (tremor, in-slab seismicity, in-slab focal mechanisms).

2) I am not a fan of the term episodic tremor and slip (ETS), especially to describe a single slow slip event. It conflates geodetic and seismic observations together, making it difficult to understand what is exactly being analyzed for an uninitiated reader, without adding any meaningful information. I would suggest referring to slow slip as simply slow slip.

Response

We are somewhat sympathetic to this opinion. However, ETS is a well-established term that highlights the central role of tremor in identifying slow slip events. Therefore, we revised our usage in the manuscript and try to limit it to places where tremor plays a key role in defining the slow slip events. At other places, in the manuscript we refer to slow slip.

3) Although the authors say that most of the identified slow slip events last for less than a week, it is important to present the context of the distribution of slow slip durations. Given that the relative timing of the stress field and b-value changes is plotted with respect to the middle of the slow slip event, it is important to know just how long these events last.

Response

Thank you for this suggestion. We added Figure S1 to show the typical duration of our slow slip events. We assessed that by stacking the numbers of tremor relative to the center of the tremor activity, defined as time 0. Figure S1 in the supplement shows that the typical duration of slow slip as detected from tremor is about 8 days, which is about 4% of the typical recurrence interval (5.6 months). By the way, slow slip events in Regions A, B, and C had very similar durations when stacked separately. Durations of the individual SSEs were also assessed by inspection of the tremor, and range from 3 to 15 days with a mean duration of 7.8 days, very similar to the result from stacking. These results are reported in the text.

4) On a similar note, why is the relative timing centered on the slow slip events? Plotting relative to a distribution of slow slip durations, the middle of the slow slip event is not a very robust reference to understand what happens before and after slow slip events. I would think it would be more much straightforward to plot everything with respect to the

start of the slow slip event. That way everything before that reference time is by definition not impacted by ongoing slip. A separate relative time scale could be created for everything after the end of all slow slip events to explore what happens after slow slip.

Response

Ideally, one would be able to clearly define the start and end of each SSE from tremor. However, with these tremor data, finding the beginning of the ETS is not always clear, whereas the middle of the event can be defined more robustly (Fig. S1). In the case of this tremor data, specifying the start and end of the event would not be justified (note that Nankai tremor data are typically sparser than in Cascadia). Also, additional complexity arises from the propagation of the SSEs – the initiation time of the ETS is well before the start of slip at a spot farther along strike. Ideally, one could ‘deconvolve’ this behavior as in Houston (2015) and determine the start time of tremor and slip at every location, but that is not feasible with these data, nor does it seem necessary here because the in-slab events with focal mechanisms can be characterized as clearly before or after the associated SSE.

5) The authors discuss fluid injection prior to slow slip from the oceanic crust into the plate interface. This does not seem very reasonable given the extremely slow production rates of fluids from metamorphic dehydration reactions (cf. Peacock et al., Geology, 2011, High pore pressures and porosity at 35 km depth in the Cascadia subduction zone) and the relatively short recurrence of slow slip in Nankai. Is a more reasonable suggestion that it is cycling of fluid pressures rather than fluid flowing itself?

Response

We agree and revised the manuscript to refer to fluid pressure changes in addition to possible fluid injections.

6) line 189-190 states that eventually the stress rotates back to its original state. Wouldn't the authors be able to estimate the duration of this healing process by extending their stress inversions to seismicity following slow slip?

Response

We did invert the focal mechanisms after the slow slip events, but a big challenge in achieving better resolution of the duration of healing is the small number of available focal mechanisms. In the supplement, we added a figure (Fig. S10) to more clearly show results of our stress inversions using 6 moving windows of 1.5 months, but the duration of the healing process remains somewhat ambiguous (Figs. S8, S9 and S10).

7) There is quite a bit text in the supplementary materials. I believe that most of this (if not all) could be incorporated into the main text and the methods section. Having as much information as possible in one document would be ideal.

Response

Thank you for this suggestion. We moved Fig. 3 (in the present manuscript), Texts S4 and S5 into the Main text (Lines 96-104) and Methods sections (Lines 254-264), respectively. On the other hand, we feel that the detailed results described in Texts S1, S2, and S3 can remain in the Supplement for optimal flow and readability.

8) *As noted above, ref. 2 presents similar observations and a similar discussion to this manuscript, yet there is no mention of it past the introduction. To provide further context, I would think there should be further discussion of this paper.*

Response

Thank you for the suggestion. We added a paragraph discussing the similarities and differences between our approach and that of Ref. 2 (Lines 110-117), and the implications of their results for our interpretation of the role of fluids.

9) *Many of the lines/symbols in the figures are not defined in the captions, e.g. Fig. 3 panel a: are the dashed lines the pre-slow slip trend? The authors should carefully review the figures and make sure that everything is accounted for in the captions or legends.*

Response

Thank you. We reviewed and clarified the figure captions, especially that of the above-mentioned figure, which is now Figure 4.

Reviewer #2 (Remarks to the Author):

This study clarified spatio-temporal correlation between temporal changes in seismicity rate, b-value, and stress orientation and episodic tremor and slip (ETS) occurring deep in the seismogenic zone using earthquake data observed around the Kii Peninsula in the Nankai Trough, and has discussed possible causes of these variations. The method used to determine the seismic source, seismicity rate, b-value, and stress orientation is well established and the results obtained have been precisely evaluated. The results of this study indicate a possibility of discovering plate slip behavior caused by ETS, which was previously overlooked, and the results of this analysis deserve to be published in Nature communication.

On the other hand, in the section discussing the causes of the spatio-temporal changes, there were some jumps in logic and insufficient explanations, probably due to the limited number of words. As mentioned below, my main comments are about the discussion, and these are mainly about descriptions, which the authors could easily correct.

Line 98-99: "The inferred asperities of anticipated great earthquake" cited here was proposed in 2003 for a disaster prevention purpose, which may not be suitable for discussing spatio-temporal changes in ETS and plate slip behavior as shown in Fig. 1. It may be more appropriate to use the results of a recent GNSS-A study on the slip deficit rate.

Response:

Thank you for your comment. We changed the figure to use slip-deficit rate contours from a recent GNSS-A study [Nishimura et al. 2018] as a proxy for the location of asperities of the anticipated great earthquake (Fig. 1a).

Line 117-118: In Fig. 3, the solid line showing the cumulative number of earthquakes after ETS is roughly parallel to the dotted line, which indicates the seismicity rate is the same as 1-3 months before ETS, although it is written as "... (seismicity rate?) decrease after ETS times" here.

Response:

We modified the text to be more accurate as the reviewer pointed out (lines 139,142-143, 521 and 523-524).

Line 128-139: The interpretation of the temporal variation of seismicity rate and b-values is mainly discussed based on the observations after the Tohoku earthquake. However, my understanding is that the Tohoku earthquake study referred to here discusses a possibility of fluid migration as an "interpretation" of the observed seismicity rate. Referring to the Tohoku study is only showing an example of a previous study that obtained similar observation, and may not appropriate as a discussion of "a causes" of the observed variation in seismicity rate and b-value. If the Swiss water injection study shows quantitative variations in seismicity and b-value with fluid injection, it would be more appropriate to precisely describe the results of the Swiss water injection study.

Response:

Thank you for your comment. We therefore added a brief description of the results of the Swiss water injection study (Lines 154-156).

Line 148-158: It seems to me the arguments here somehow farfetched. For example, this paragraph mentions that "the formation of quartz veins is generally associated with hydrothermal circulation", therefore, "the quartz vein sealing cycle is related to the in-slab seismicity". To me there are some jumps of logic between the two sentences. More explanation may be necessary. And also, it seems to me that this paragraph mentions that the reference 21 and 22 concluded that "low attenuation layer" = "impermeable layer". Do the reference 21 and 22 conclude so?

Response:

Thank you for this comment. We reorganized and rewrote this paragraph (Lines 175-188) to clarify and strengthen our line of reasoning. We now cite Kita and Matsubara, 2016 (Ref. 17) as the attenuation reference in line 177. This reference found a low-attenuation zone above the slab, interpreted as a dry zone. The current manuscript then interprets that lack of fluid as due to a thin layer of impermeable quartz, predicted to be present from geological observations. We further clarified the explanation-by indicating that the proposed relationship between the quartz vein sealing cycle and in-slab seismicity is our interpretation rather than an observation. The paragraph (Lines 175-188) was modified to reflect these points.

Line 196-197: I think this sentence is an exaggeration. The approach by this study is useful for understanding slip phenomena after the events, but it may not be suitable for "prediction" because this study's approach seems to require "retrospective" analysis of earthquake data "after" the event occurs.

Response:

Thank you for this feedback. We modified the sentence to: 'The proposed simple methodology could contribute to better understanding for the nucleation of future megathrust earthquakes' (Lines 226-227).

REVIEWERS' COMMENTS

Reviewer #1 (Remarks to the Author):

I thank the authors for the detailed responses to both my own comments and those of the other reviewer. I continue the below discussion on the specific points raised in my previous review (if not mentioned, then I consider that comment addressed!). I think some minor changes are still necessary, but they can perhaps be addressed during proofs rather than going through another round of revision/reviews.

2) I would strongly push back against referring to slow slip in two different ways: ETS and slow slip. This creates the impression for readers that there are two different phenomena, when in reality both refer to aseismic fault slip on the plate interface. There is of course some slow slip that is accompanied by tectonic tremor and LFEs and others that are not, but making this distinction between ETS and "slow slip" implies that there are mechanical differences between the two. It can simply be made clear that tremor here is used as a proxy to define when slow slip is occurring (which is well justified from many recent papers), e.g. "We define here a slow slip event from the spatiotemporal record of tremor activity." There is also the added benefit of having one less acronym to obfuscate the manuscript. Simplification of the jargon is always good, especially when the jargon does not provide any useful precision.

4) Looking at Figure S1, it would seem straightforward to define an arbitrary threshold to determine the start and end of each slow slip event. That said, the duration of slow slip is likely too short to be of use when estimating the properties of the stress field over week- or month-long time scales. So I suppose this is a moot point. Thank you for the clarifying figure!

6) Why not go beyond 3 months in Figure S10? I suppose it looks from figure 1 that 3 months is the average recurrence interval...? I wonder if it wouldn't be useful to have a supplementary figure that tracks the evolution of the stress field over the entire dataset (even if the results are clearest when stacking on times of slow slip). This would provide useful context for the results stacked on times of slow slip.

Reviewer #2 (Remarks to the Author):

The authors responded appropriately to all the comments from the reviewers, and I was satisfied with their responses. I think that the revised manuscript is ready for publication with Nature Communications.

Response to Reviewers

We thank the Reviewers for their helpful comments which have improved the clarity of our manuscript. We moved one figure into the main manuscript from the supplemental materials. We also changed description about aseismic slips.

Reviewer #1 (Remarks to the Author):

Comments:

2) I would strongly push back against referring to slow slip in two different ways: ETS and slow slip. This creates the impression for readers that there are two different phenomena, when in reality both refer to aseismic fault slip on the plate interface. There is of course some slow slip that is accompanied by tectonic tremor and LFEs and others that are not, but making this distinction between ETS and "slow slip" implies that there are mechanical differences between the two. It can simply be made clear that tremor here is used as a proxy to define when slow slip is occurring (which is well justified from many recent papers), e.g. "We define here a slow slip event from the spatiotemporal record of tremor activity." There is also the added benefit of having one less acronym to obfuscate the manuscript. Simplification of the jargon is always good, especially when the jargon does not provide any useful precision.

Response

Thank you for your comment. We used ETS because many papers used ETS for southwestern Japan and Cascadia. We can appreciate the Reviewer's point of view. But we need a term to distinguish between the region hosting recurrent large 30-40 km deep slow slip events (that form the basis of our analysis) and the region updip of there, which we infer from our stress change results also hosts some slow slip but without tremor. Therefore, we adopt the term "slow slip with tremor" (SST) to emphasize that tremor is part of the process of slow deformation on the deep plate interface, and refer to the SST zone and the updip zone. We added a few sentences of explanation to the main text.

4) Looking at Figure S1, it would seem straightforward to define an arbitrary threshold to determine the start and end of each slow slip event. That said, the duration of slow slip is likely too short to be of use when estimating the

properties of the stress field over week- or month-long time scales. So I suppose this is a moot point. Thank you for the clarifying figure!

Response

Yes, unfortunately, obtaining more temporal detail of the stress evolution is limited by the fairly small number of in-slab events with focal mechanisms.

6) Why not go beyond 3 months in Figure S10? I suppose it looks from figure 1 that 3 months is the average recurrence interval...? I wonder if it wouldn't be useful to have a supplementary figure that tracks the evolution of the stress field over the entire dataset (even if the results are clearest when stacking on times of slow slip). This would provide useful context for the results stacked on times of slow slip.

Response

Thank you for the suggestion. Obtaining more temporal detail of the stress evolution is limited by the fairly small number of in-slab events with focal mechanisms. However, prompted by the Reviewer's comments, we have added results of inversions of two more periods to Figure S10: -3.0 to -1.5 months and 1.5 to 3.0 months.